

# Software Program Development of a High-Precision Magnetometer System for Human-occupied Vehicles

Qimao Zhang[1,2], Keyu Zhou[1], Ming Deng[1], Ling Huang[2], Cheng Li[2] and Qisheng Zhang[1]

[1]School of Geophysics and Information Technology, China University of Geosciences (Beijing), Beijing 100083, China

[2]Aerospace Information Research Institute, Chinese Academy of Sciences, Beijing 100190, China

*Correspondence to*: Ming Deng (dengming@cugb.edu.cn)

**Abstract.** Cesium optically pumped magnetometers are widely used in geophysical exploration, environmental monitoring, and scientific research for magnetic field measurements. Traditional magnetometer systems, however, encounter limitations in complex environments due to magnetic interference, low data acquisition efficiency, and inadequate real-time control, which

affect both application and accuracy. To overcome these challenges, this paper presents a software system designed for cesium optically pumped magnetometers. The system incorporates automatic probe switching, data acquisition, user interface control, and a compensation algorithm. The automatic switching function mitigates dead zones in the optically pumped probes, ensuring continuous data collection. The data acquisition module supports multiple formats and transmission protocols, enhancing efficiency and ensuring data integrity. The user interface facilitates real-time monitoring and control of

magnetometer operations. The system employs the Tolles-Lawson model to effectively suppress environmental magnetic interference, ensuring high-precision measurements even in complex environments. Experimental results confirm the system's enhanced sensitivity and stability, making it a reliable tool for precise magnetic field measurements in challenging conditions.

## 1 Introduction

High-precision magnetic field measurements have significant applications in various fields such as geophysical exploration

(Outlook on the Worldwide Quantum Magnetometer Industry to 2029), environmental monitoring (Tang et al., 2023), archaeology (Eppelbaum and Mishne, 2011; Stele et al., 2023; Schmidt et al., 2024), and scientific research (Bennett et al., 2021). As a crucial measurement tool, magnetometers are capable of accurately detecting the Earth's magnetic field, underground mineral deposits, and environmental changes. However, traditional magnetometer systems face numerous challenges in practical applications, such as environmental magnetic interference, low data acquisition efficiency, and

insufficient real-time control capabilities. These issues limit their application and measurement accuracy in complex environments. In particular, environmental magnetic interference and noise significantly affect measurement accuracy, making it imperative to effectively eliminate these interferences. Additionally, traditional data acquisition software often struggles with processing large volumes of data efficiently, compromising data integrity and reliability. The lack of real-time control capabilities further hinders users from making quick adjustments and optimizations during measurements.





Cesium optically pumped magnetometers, known for their high sensitivity and precision, have become essential instruments in modern magnetic field measurements (Lebedev et al., 2020). To address the aforementioned challenges, researchers have proposed various improvements in recent years (Lebedev et al., 2020; Dong et al., 2021). However, most studies have focused on hardware enhancements (Xue et al., 2021; Teng et al., 2024; Liu et al., 2018; Liu et al., 2020; Dou et al., 2018; Peng et al., 2021; Schultze et al., 2015; Greene et al., 2024), with relatively few addressing software systems (Hu et al., 2017; Narita et al., 2021), particularly in terms of improving data acquisition efficiency and real-time control capabilities. In fact, software systems play a crucial role in enhancing the efficiency and accuracy of magnetometer measurements. Efficient data acquisition software can significantly increase data processing speed, ensuring data integrity and reliability. Powerful display and control software can provide intuitive user interfaces and flexible control functions, enhancing user experience. Advanced compensation algorithms can effectively eliminate environmental interferences, improving measurement accuracy.

To improve the measurement accuracy and application efficiency of cesium optically pumped magnetometers in complex environments, this paper designs and implements a high-precision magnetometer software system based on the HOV (Human Occupied Vehicle) platform. The system primarily consists of probe automatic switching software, data acquisition and processing software, and upper computer software. The probe automatic switching system is used to automatically switch between three working probes, avoiding probe dead zones and enhancing measurement continuity and reliability. The data acquisition and processing software is responsible for efficient data collection and storage and includes real-time magnetic compensation data processing functions. By introducing the Tolles-Lawson model, it effectively eliminates environmental magnetic interference. The upper computer main control system runs the display and control software, providing an intuitive user interface and powerful control functions, allowing users to monitor and adjust the magnetometer's operational status in real-time. Rigorous testing has demonstrated that the probe achieves noise levels as low as 1 pT/√Hz, with the system exhibiting a sensitivity of 0.5 nT. Under the monitoring and control of the software, the system has consistently demonstrated its capability to accurately collect data while maintaining stable and reliable operational performance.

## 2 Overall Design of the High-Precision Cesium Optically Pumped Magnetometer Based on the HOV Platform

The block diagram of the high-precision cesium optically pumped magnetometer system proposed in this paper is shown in Figure 1. It consists of three main parts: the front-end sensor section, the data acquisition and processing system, and the upper computer main control system.

The front-end sensor section includes a set of optically pumped sensors and magnetic compensation probes, which convert magnetic field signals into electrical signals and transmit them to the subsequent acquisition board. The data acquisition and processing system is composed of filter circuit modules, amplification circuit modules, waveform shaping modules, temperature control modules, RF excitation circuit modules, power supply modules, and the main control processor. The main control processor, centered around the Zynq platform, integrates an analog-to-digital conversion (ADC) chip, comparator





shaping circuits, data interface circuits, clock circuits, and power modules. It is responsible for converting the incoming analog signals into digital signals and communicating with the upper computer on the HOV platform.

The upper computer main control system communicates with the data acquisition and processing system via Gigabit Ethernet, allowing users to monitor and adjust the magnetometer's operational status in real-time.

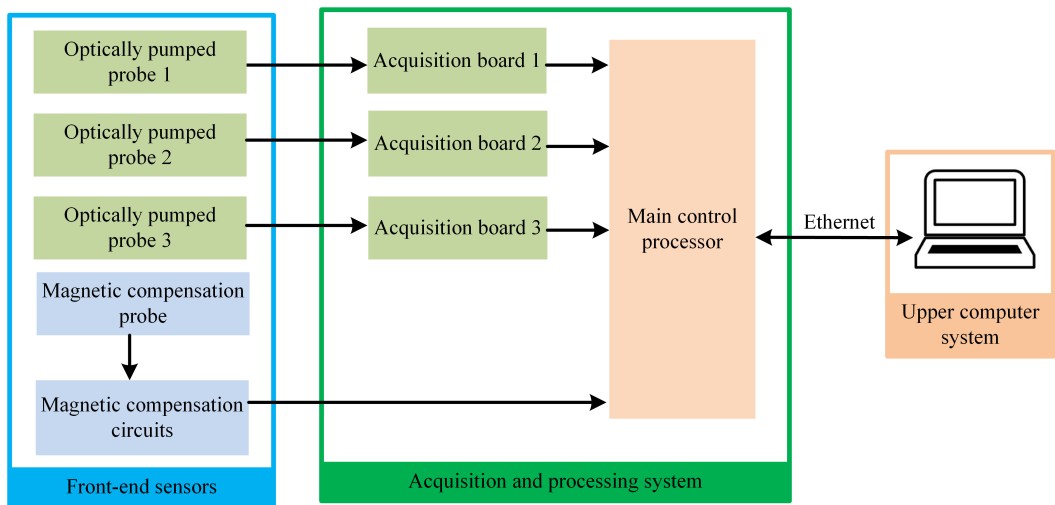


**Figure 1: Block diagram of the cesium optically pumped magnetometer system.**

## 2.1. Hardware platform

The block diagram of the main control processor is shown in Figure 2. Two types of analog signals from the front-end acquisition section enter the processor. The signal from the magnetic compensation probe first passes through an impedance

matching circuit to obtain the most accurate analog signal. Next, an anti-aliasing low-pass filter is designed according to the sampling rate. Finally, the signal is converted to a digital signal through an ADC.

The Zynq PL (Programmable Logic) side controls the ADC chip for data acquisition while temporarily storing the collected data in the system memory. The Larmor frequency signal from the optically pumped probe is coupled to the power line output through a transformer, with a frequency range of 50 kHz to 350 kHz and an amplitude of approximately 300 mVpp. First, the

Larmor frequency signal is separated from the power line. Then, it undergoes low-pass filtering to remove out-of-band noise. Finally, an ultra-high-speed hysteresis comparator outputs a low-jitter digital logic signal. The PL side calculates the frequency of the digital logic signal and outputs the magnetic field value. A temperature-compensated crystal oscillator provides a high-precision clock source for frequency measurement. The PS (Processing System) side mainly stores the data to a local TF card and communicates with the upper computer for real-time data transmission.





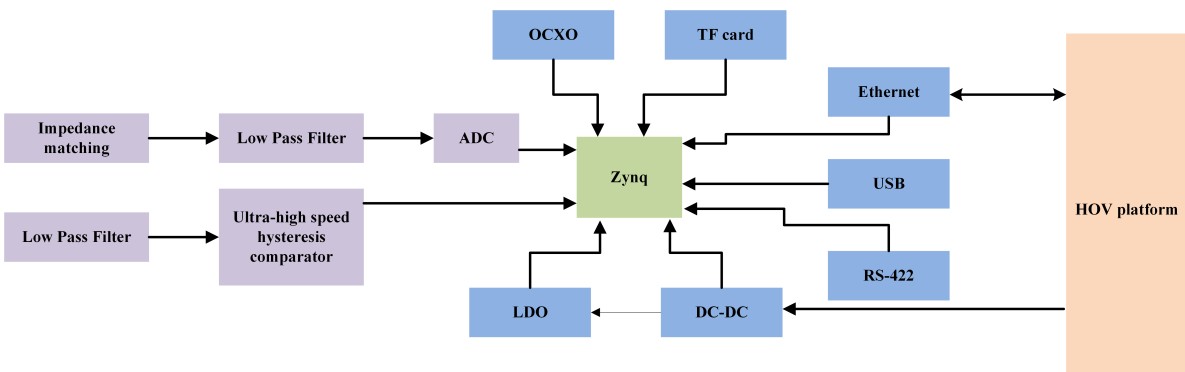


**Figure 2: Block diagram of the main control processor.**

## 2.2. Software Overall Design

The software components are mainly integrated into the data acquisition and processing system and the upper computer main control system. The data acquisition and processing software system includes probe automatic switching software, data

acquisition software, real-time magnetic compensation data processing software, data storage software, and data communication software. The upper computer main control software system comprises the data communication module, display module, and data storage module. The overall software architecture is shown in Figure 3.

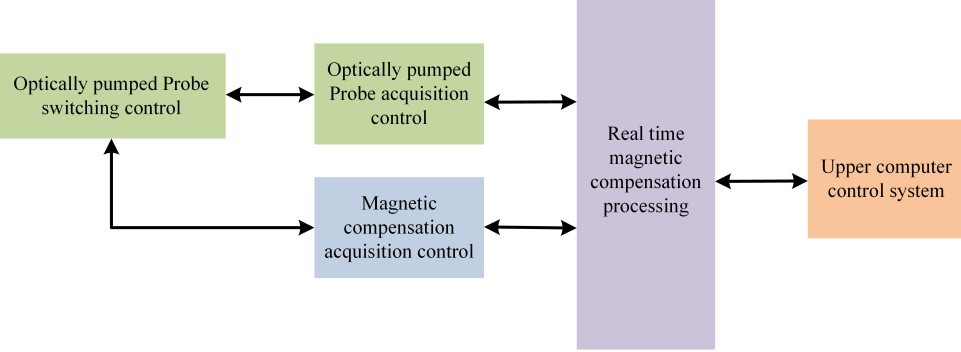

**Figure 3: Overall software architecture.**

**3 Design of Probe Automatic Switching Software**

Optically pumped probes possess high sensitivity and precision in magnetic field measurements. However, when the magnetic field direction is at 90° or 0° to the probe's optical axis, the probe's sensitivity is at its lowest, creating dead zones. These dead zones can lead to decreased measurement accuracy, data discontinuity, and limited application range (Zhou et al., 2024). To address this issue, we employ a multi-probe system and automatic switching software. By using multiple optically pumped

probes installed at different angles to cover various magnetic field directions, and designing an automatic switching software, the system can automatically select the optimal working probe based on the current magnetic field direction. This approach

avoids the dead zones of the probes, ensuring continuous and reliable measurements. The hardware setup of the three-probe system is shown in Figure 4.

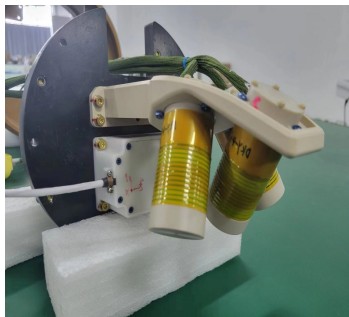

**Figure 4. Hardware setup of the three-probe system.**

The process of the probe automatic switching software is shown in Figure 5. First, the magnetic compensation probe is used to calculate the current magnetic inclination (±90°) and magnetic declination (±180°). Then, combined with the known installation angles of the three optically pumped probes relative to the magnetic compensation probe, the angle between each optically pumped probe and the geomagnetic field is calculated. The specific steps include: obtaining the measurement results

from the magnetic compensation probe, determining the installation angles of the optically pumped probes, and calculating the actual angle between each probe and the geomagnetic field through geometric relationships. Ideally, the working angle is optimal when the angle between the optically pumped probe and the geomagnetic field is 45°. Therefore, the next step is to calculate the actual angle between each probe and the geomagnetic field and compare it with 45° to obtain the working deviation value for each probe. By comparing the deviation values of the three probes, the probe with the smallest deviation is

selected as the current working probe. In this way, the probe automatic switching software can intelligently select the optimal working probe, avoiding the dead zones of the probes and ensuring continuous and accurate measurements.

To further enhance the reliability and practicality of the system, the probe automatic switching software records the time, reason, and selected probe for each switching event, facilitating subsequent analysis and optimization. Additionally, the upper computer software provides an intuitive user interface that displays the current magnetic field measurements, probe status, and

switching records, allowing users to monitor and operate the system in real-time. Finally, the system includes an alarm mechanism that can issue alerts in the event of probe switching failures or abnormal magnetic field conditions, prompting users to check and address the issues.





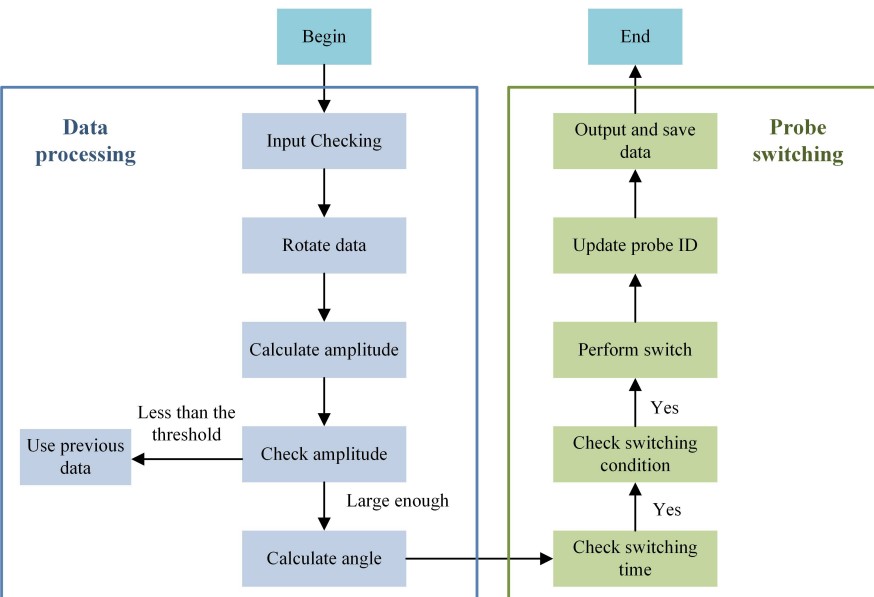

**Figure 5. Flowchart of the probe automatic switching software.**

## 4. Design of the Data Acquisition and Processing Software System

Based on the application requirements of the carrier magnetic interference compensation system, the software system includes the following functions:

1.        Data Communication Function: Establishes network communication with the display and control computer, enabling command parsing and data uploading.

2.        Data Collection and Control Function: Collects data from the optically pumped magnetometer, three-axis fluxgate magnetometer, and auxiliary information, while also controlling hardware operational states and parameter settings.

3.        Real-time Processing of Magnetic Compensation Data: Utilizes magnetic compensation coefficients to perform real-time calculations to compensate for magnetic interference effects on the HOV platform.

4.        Data Storage Function: Stores navigation state parameters, compensation coefficients, raw magnetic data, and processed results during the survey.

The system reports real-time data every 100ms. The raw magnetic field data is collected at 120Hz and downsampled to a 100ms interval through filtering. All magnetic compensation data processing and magnetic anomaly detection are performed using the 100ms interval data. The real-time data reported includes raw geomagnetic data, raw magnetic compensation data, and compensated geomagnetic data, all downsampled to a 100ms interval through filtering. The external communication interface uses a network interface to communicate with the upper computer main control system, and data acquisition utilizes RS422-based serial communication. Table 1 lists the external interfaces of the data acquisition and processing software.



**Table 1.** External interfaces of the data acquisition and processing software.

| Interface | Description | Interface Type | Trigger Mode | Sender | Receiver |
|---|---|---|---|---|---|
| NET Communication Interface | The HOV platform sends time, carrier status information, and control commands to the data acquisition and processing software. | Serial port Data Packet | Scheduled Trigger | HOV Platform | Data Acquisition and Processing Software |
| | The data acquisition and processing software sends real-time magnetic field data, magnetic anomaly target information, device self-check information, and control command feedback information to the HOV platform. | Serial port Data Packet | Scheduled Trigger | Data Acquisition and Processing Software | HOV Platform |
| RS422 Data Acquisition Serial Port | The optically pumped probe electronic unit sends the raw magnetic field data to the data acquisition and processing software. The magnetic compensation data and control command feedback are sent back to the data acquisition and processing software. | Serial port Data Packet | Command Trigger | Optically Pumped Probe Electronic Unit | Data Acquisition and Processing Software |
| | The data acquisition and processing software sends control commands to the optically pumped probe electronic unit. | Serial port Data Packet | Command Trigger | Data Acquisition and Processing Software | Optically Pumped Probe Electronic Unit |

Each functional module of the data acquisition and processing software operates independently and uses multithreading for
simultaneous processing. Data transfer between functional modules is achieved through multi-level shared buffers. Shared buffers are created for five internal interfaces, and each functional module retrieves data from the circular buffer according to the interface-defined structure, processes it, and then writes the processed data back into the input circular buffer. The data storage module operates as a separate thread, reading data from the multi-level buffers at fixed intervals and writing it to files. The internal interface objects of the data acquisition and processing software include: the data acquisition and control module,
the magnetic compensation coefficient invocation module, the magnetic compensation data processing module, the device self-check and status reporting module, and the data storage module. The internal interface diagram of the software is shown in Figure 6, and the internal interface list is presented in Table 2.





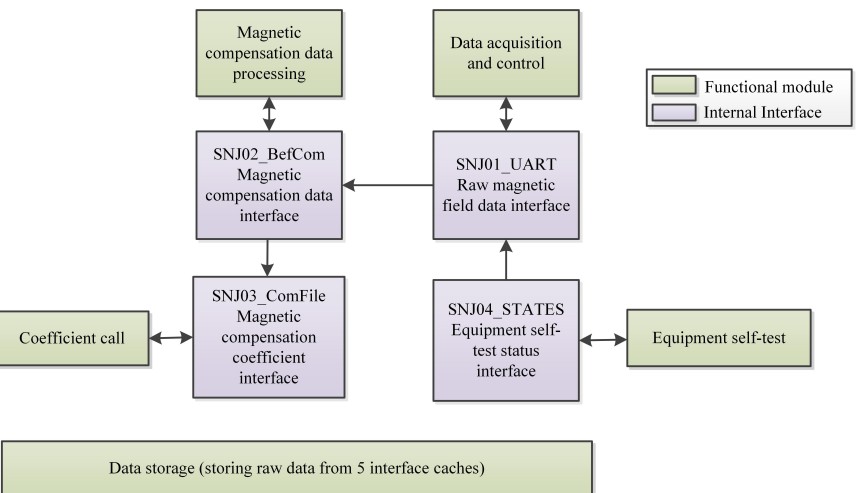

**Figure 6: Internal interface diagram of the data acquisition and processing software.**

**Table 2.** Internal interface list of the data acquisition and processing software.

| Interface | Description | Interface Type | Trigger Mode | Sender | Receiver |
|---|---|---|---|---|---|
| Raw Magnetic Field Data Interface | The data acquisition and control module parses the raw data collected from the optically pumped probe electronic unit to generate initial magnetic field data. | Buffer Data Packet | Scheduled Trigger | Serial port raw data | Data Acquisition and Control Module |
| Magnetic Compensation Data Interface | The data acquisition and control module processes the initial magnetic field data by downsampling and filtering to generate downsampled and filtered raw magnetic field data. | Buffer Data Packet | Scheduled Trigger | Data Acquisition and Control Module | Magnetic Compensation Data Processing Module |
| Magnetic Compensation Coefficient Interface | The magnetic compensation coefficient invocation module retrieves and calculates the magnetic compensation coefficients. | File | Command Trigger | Emergency Transmission Module | Data Storage Module |
| Device Self-Check Status Interface | The data acquisition and control module retrieves device self-check status from the optically pumped probe electronic unit based on self-check commands. | Data Packet | Command Trigger | Data Acquisition and Control Module | Device Self-Check Status Module |



## 4.1. Data Acquisition and Control Software

The data acquisition and control module primarily implements the functions of magnetic compensation probe data acquisition
control, optically pumped probe data acquisition control, hardware status monitoring, self-check, and fault detection. The data
acquisition and control module collects raw magnetic field data from the magnetic compensation probe and the optically
pumped probe through RS422 serial port. Additionally, it merges the current system status, including latitude, longitude,
altitude, and control status words, into the data to generate raw magnetic detection data, which is then written into the data
buffer. The logical flowchart is shown in Figure 7.

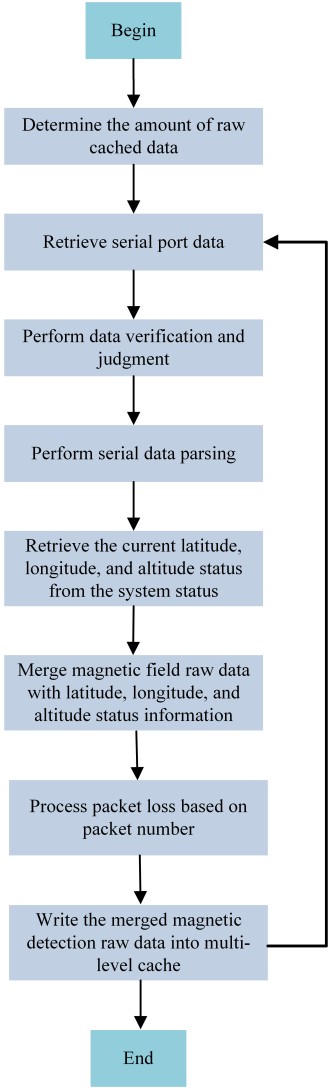


**Figure 7: Logical flowchart of the data acquisition module processing.**





Exception Handling: Initially, the raw serial port data undergoes frame header and frame trailer verification to ensure data integrity. Subsequently, an XOR checksum is performed on the raw serial port data to prevent the propagation of erroneous data to subsequent processing stages in the presence of interference. Data that fails the checksum is treated as packet loss.

Moreover, for abnormal data collected from the serial port, spike removal processing is employed. This involves defining threshold limits for sudden anomalous values, such as those induced by engine startup interference or other equipment power-on interference. Data exceeding these limits is replaced with the preceding data point.

For raw serial port data, packet loss detection is implemented. If the packet loss duration is less than 5 seconds, the most recent data point is utilized as a replacement to maintain the number of sample points and ensure the temporal consistency of data

processing. In instances where the packet loss duration exceeds 5 seconds, the data acquisition process is restarted. This methodology ensures the robustness and reliability of the data acquisition system, mitigating the impact of transient errors and preserving data integrity.

**4.2. Magnetic Compensation Data Processing Software**

The Tolles-Lawson model (Feng et al., 2022) is a mathematical model used for magnetic compensation and is widely applied

in fields such as aeromagnetic surveys, marine magnetic surveys, and terrestrial magnetic surveys. This model was proposed by R. H. Tolles and C. A. Lawson in the 1960s with the aim of compensating for various interference factors in magnetic field measurements, particularly errors caused by the inherent magnetism and electromagnetic interference of the carrier (e.g., aircraft or ships) (Hezel, M.C., 2020).

According to the Tolles–Lawson model, the interference magnetic field generated by the carrier platform can be divided into

three types: constant magnetic field, induced magnetic field, and eddy current magnetic field. If $H_t(t)$ is defined as the total intensity of the interfering magnetic field, then:

$$H_t(t) = H_{CONS}(t) + H_{IND}(t) + H_{EDDY}(t) \tag{1}$$

The constant magnetic field $H_{PERM}(t)$ caused by the inherent magnetism of the carrier ferromagnetic component can be expressed as

$$H_{CONS}(t) = c_1 u_1 + c_2 u_2 + c_3 u_3 = \sum_{i=1}^{3} A_i(t) c_i \tag{2}$$

Among them, $c_i$ is the constant magnetic field compensation coefficient, and $A_i(t)$ is a variable composed of direction cosine. The induced magnetic field $H_{IND}(t)$ can be expressed as

$$H_{IND}(t) = H_e(t)(c_4 u_1^2 + c_5 u_1 u_2 + c_6 u_1 u_3 + c_7 u_2^2 + c_8 u_2 u_3 + c_9 u_3^2)$$
$$= \sum_{i=4}^{9} A_j(t) c_j \tag{3}$$





Among them, $c_j$ is the compensation coefficient of the induced magnetic field, and $A_j(t)$ is the variable composed of the environmental magnetic field and the direction cosine.

The eddy current magnetic field $H_{EDDY}(t)$ can be expressed as

$$
\begin{aligned}
H_{EDDY}(t) = H_e(t)(&c_{10}u_1u_1' + c_{11}u_1u_2' + c_{12}u_1u_3' + c_{13}u_2u_1' \\
&+ c_{14}u_2u_2' + c_{15}u_2u_3' + c_{16}u_3u_1' + c_{17}u_3u_2' + c_{18}u_3u_3') = \sum_{i=10}^{18} A_k(t)c_k
\end{aligned}
\tag{4}
$$

Among them, $u_1'$, $u_2'$, and $u_3'$ are the derivatives of $u_1$, $u_2$, and $u_3$, respectively. $c_k$ is the compensation coefficient of the

eddy current magnetic field, $A_k(t)$ is a variable composed of the environmental magnetic field $H_e(t)$ and the direction cosine. Corresponding matrix is as follows:

$$
A = \begin{bmatrix}
A_1(1) & A_2(1) & \cdots & A_{18}(1) \\
A_1(2) & A_2(2) & \cdots & A_{18}(2) \\
\vdots & \vdots & \ddots & \cdots \\
A_1(n) & A_2(n) & \cdots & A_{18}(n)
\end{bmatrix}
\tag{5}
$$

The compensation coefficient for interference magnetic field, which is used in figure 8, can be obtained by solving the

following formula:

$$
C = (A^T A)^{-1} A^T H_t
\tag{6}
$$

The magnetic field data processing module integrates data from both the optically pumped probe and the magnetic compensation probe to generate magnetic compensation data. The flowchart of the magnetic field data processing is shown in Figure 8.



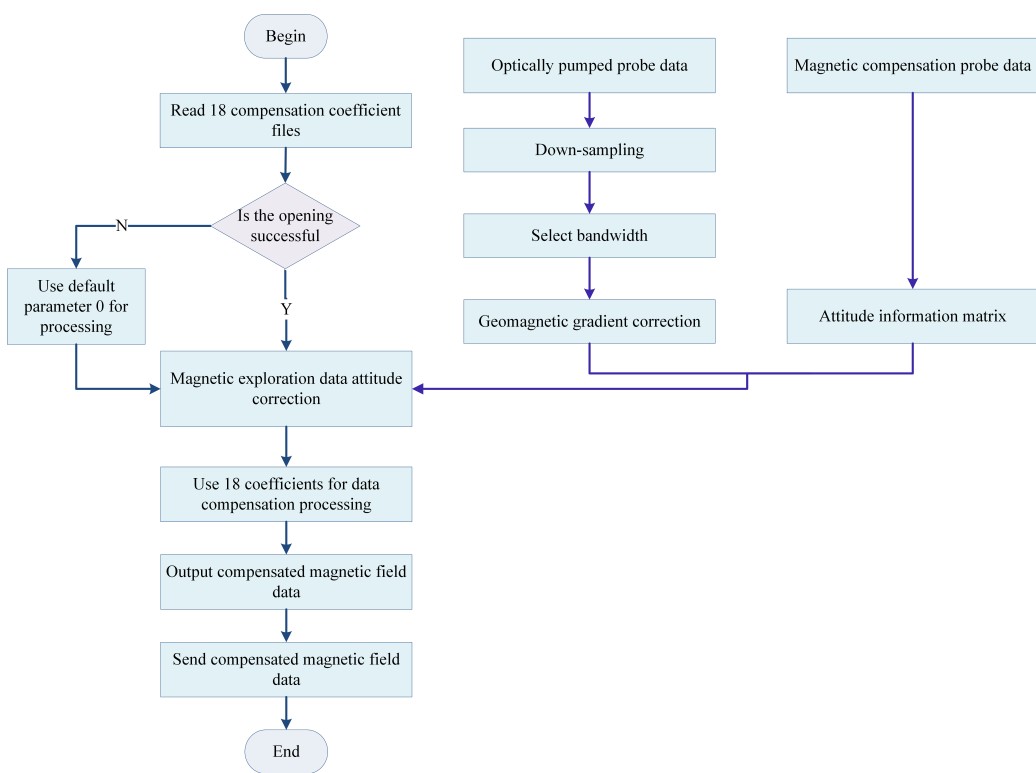


**Figure 8: Flowchart of magnetic field data processing.**

## 4.3. Magnetic Detection Status Reporting and Data Storage Function

The status reporting function module of the magnetic detection instrument is implemented through network-based data communication. During operation, the magnetometer automatically performs data acquisition upon power-up and

simultaneously reports the device status to the upper computer's magnetic measurement display and control software. The data storage module creates a data file based on the system time during software initialization. Under normal operating conditions (magnetic compensation mode, magnetic compensation with automatic identification mode, or magnetic compensation with manual identification mode), the module determines the size of unsaved data in the buffer by calculating the difference between the read and write pointers of the multi-level circular buffer at one-second intervals. It then writes the unsaved data from the

buffer to the data file. When creating a file, the system checks the remaining disk space. If the available storage space is less than 100MB, data acquisition is halted, and data storage is discontinued to prevent data loss.

## 5. Upper Computer Software Design

The upper computer utilizes Gigabit Ethernet to receive data from the magnetic detection processing software and relevant status information from the HOV platform. The software includes a communication module, a data display module, and a data





storage module. It is primarily responsible for controlling the magnetometer, configuring parameters, and real-time reception, visualization, and storage of various types of data within the system. The design interface of the upper computer software is shown in Figure 9.

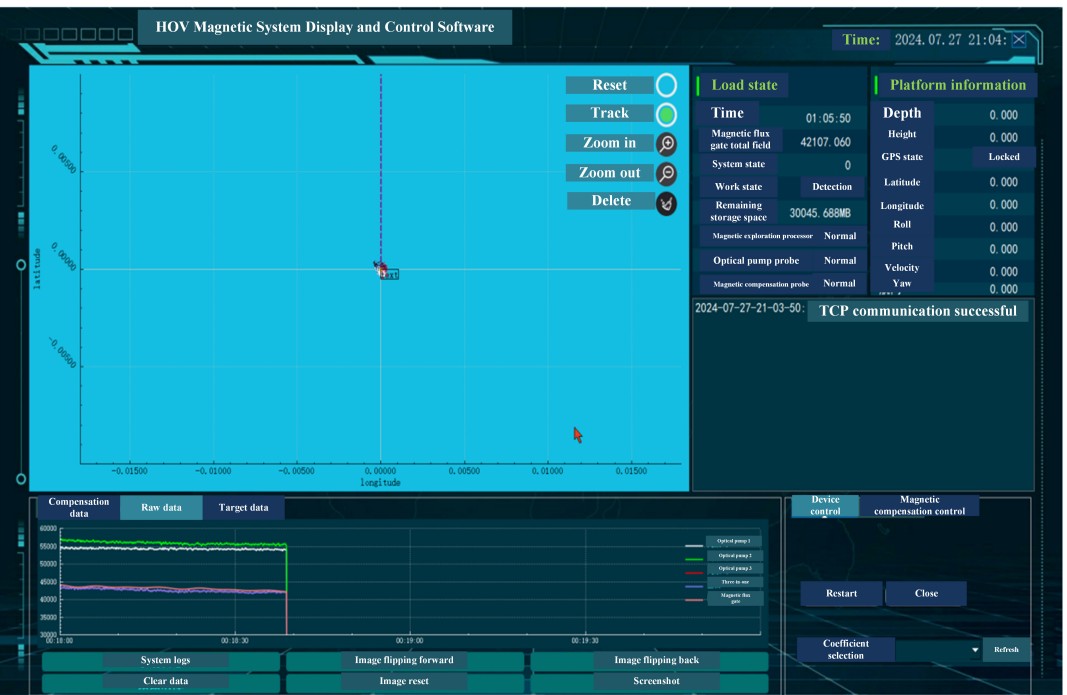

**Figure 9: Interface of the Upper Computer Software.**

**5.1. Data Reception and Storage**

The data reception module is responsible for receiving data and status information, performing data validation and parsing in accordance with the specified data protocol, and subsequently storing the parsed data in a shared buffer. The process commences with the initialization of the network connection and the binding of the reception slot function. Following this, the system receives the network protocol information pertaining to the HOV platform's system status. The received data is then

parsed according to the predefined network protocol structure and stored in the data buffer. The flowchart illustrating the data reception process is depicted in Figure 10.





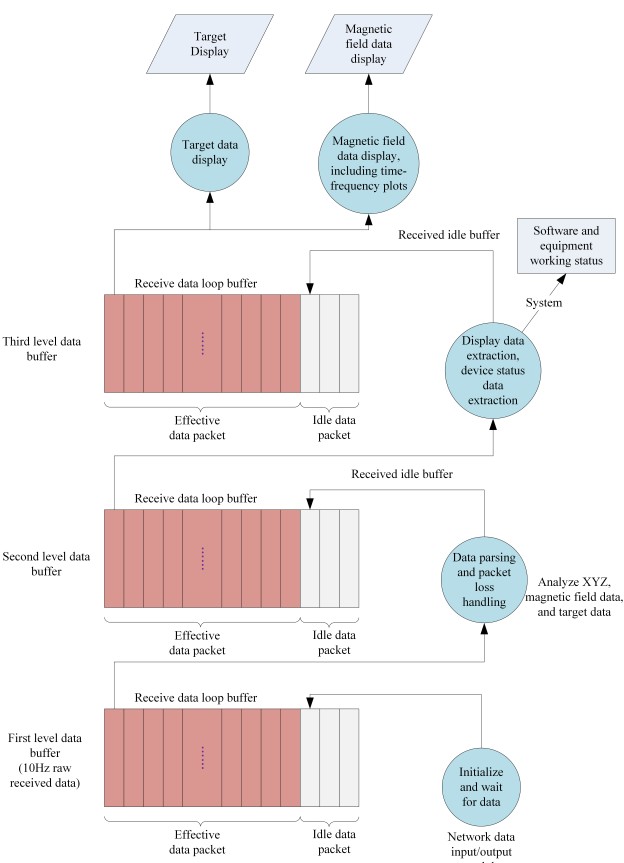

**Figure 10: Flowchart of the data reception process.**

**5.2. Image Display Processing**

The image display processing module extracts platform position information from the data buffer and performs real-time trajectory visualization. Simultaneously, it receives status information from the magnetometer equipment and magnetic measurement data. The device status bar displays the connection status of the optically pumped probe, the magnetic compensation probe, and the interface unit. The real-time graph displays both the raw magnetic measurement data and the compensated magnetic measurement data.

In the software design and development phase, the logical processing and display components are implemented separately, with communication facilitated through shared memory. The shared memory objects include raw data and magnetic compensation data. Due to filtering delays, there is a time lag between the raw data and the magnetic compensation data, necessitating synchronization based on corresponding timestamps. The synchronization method involves the processing module sending the current timestamps of the raw data and magnetic compensation data to the display module, which uses

this information to achieve synchronization. The image processing functions include vertical and horizontal axis scaling, target information scale identification, image panning, and image resetting.



## 6. System Testing

In this section, we present the testing procedures and results for the high-precision magnetometer system based on the HOV platform, as designed in this study. The objective is to validate the system's feasibility and analyze the effectiveness of the magnetic compensation.

### 6.1. Laboratory Testing

In a controlled laboratory environment, we conducted comprehensive debugging and performance evaluation of the system. The test results demonstrate that the probe noise is as low as 1 pT/√Hz, indicating exceptional sensitivity. The magnetic measurement system consistently collected accurate data and maintained stable operation. Under the supervision of the monitoring software, the system demonstrated high efficiency in executing detection tasks.

### 6.2. Marine Testing Procedure

To further evaluate the performance of the system, a marine experiment was conducted on July 28, 2024. During the TS2-38-7 expedition of the Explorer II (from July 20 to July 29, 2024), the HOV magnetometer was utilized in the SY717 dive for comprehensive system functionality verification and magnetic compensation testing. The submersible was deployed at 8:30 AM, and the equipment was powered on and checked at 8:51 AM. The submersible reached the seabed at approximately 9:18 AM at a depth of around 940 meters. At 10:09 AM, the retrieval target was located, and the cable-cutting operation was completed by 10:20 AM. Subsequently, a magnetic compensation test was conducted at an altitude of 50 meters above the seabed. The submersible jettisoned its ballast at 11:47 AM and surfaced at approximately 12:14 PM.

The overall functionality of the equipment was verified to be normal. The magnetic data acquisition and the parsing of auxiliary information (including latitude, longitude, attitude, and depth) were confirmed to be accurate. Figure 11 presents a photograph of the equipment at a depth of 940 meters. Figure 12 illustrates the 3D trajectory of the HOV as recorded by the magnetometer. The anomalies in the latitude and longitude data were attributed to the real-time positioning information of the HOV not being updated, as confirmed by on-site verification.



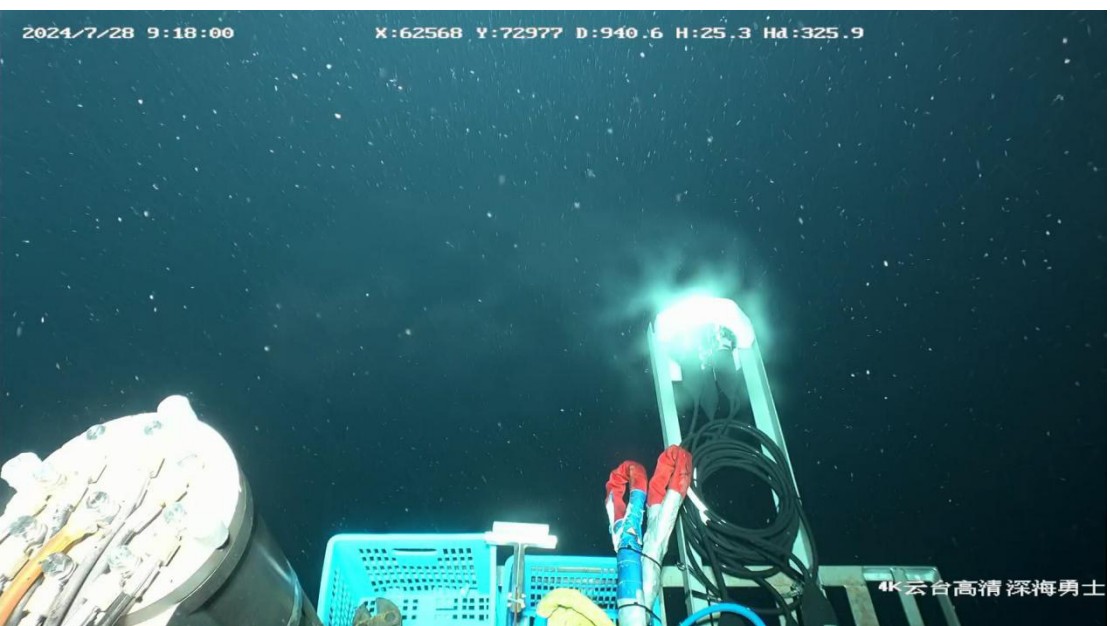

**Figure 11: Photograph of the equipment at a depth of 940 meters.**

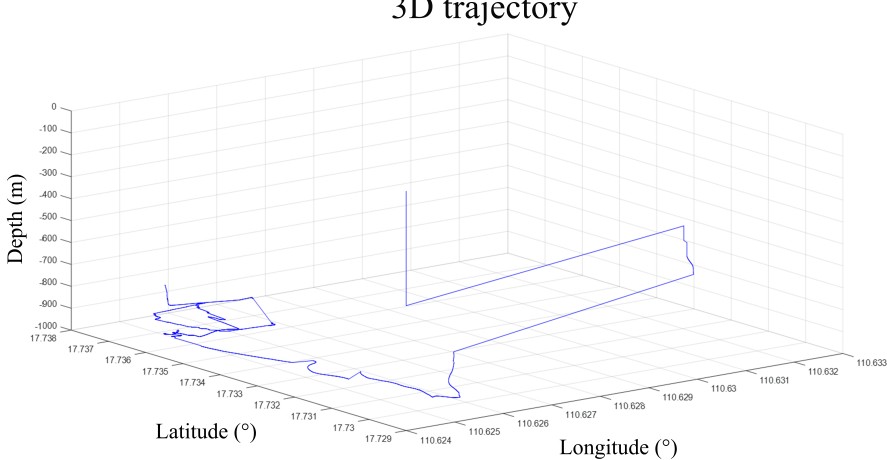

**Figure 12: 3D trajectory of the HOV recorded by the magnetometer.**

## 6.3. Analysis of Marine Experiment Results

The comprehensive magnetic measurement values and the pitch and roll data recorded by the HOV magnetometer are depicted in Figures 13 and 14, respectively. The results demonstrate that the latitude, longitude, depth, and magnetic measurement values recorded by the HOV magnetometer are consistent and accurate. The integrity of the recorded magnetic measurement





data is maintained throughout the operation. Notably, significant magnetic field variations are observable when approaching the retrieval target and during specific operational phases, indicating the system's sensitivity to magnetic anomalies.

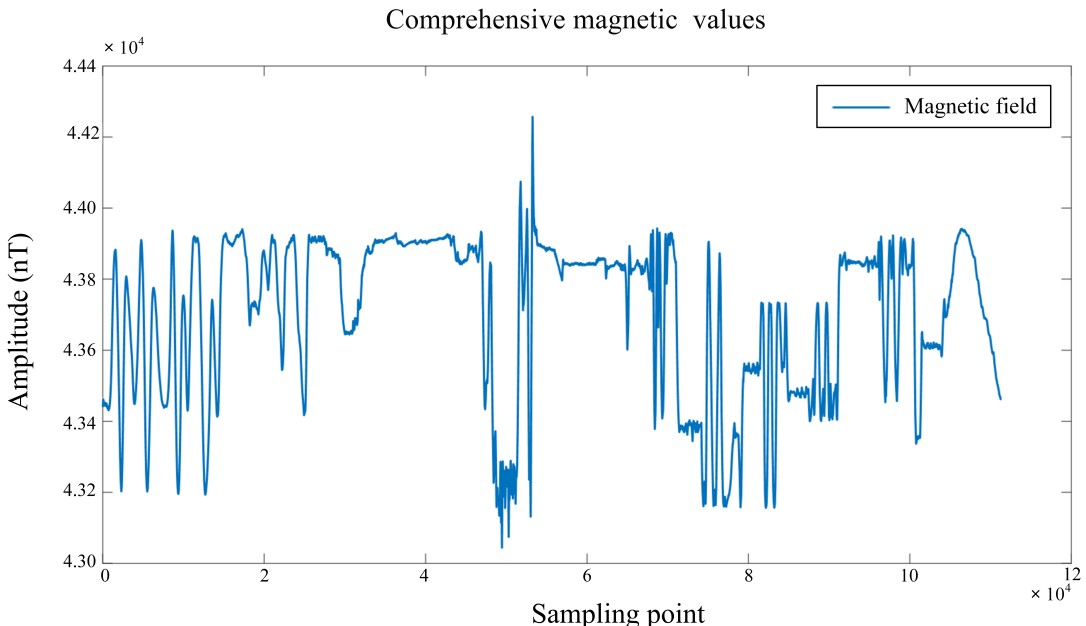

**Figure 13: Comprehensive magnetic values recorded by the magnetometer.**

280

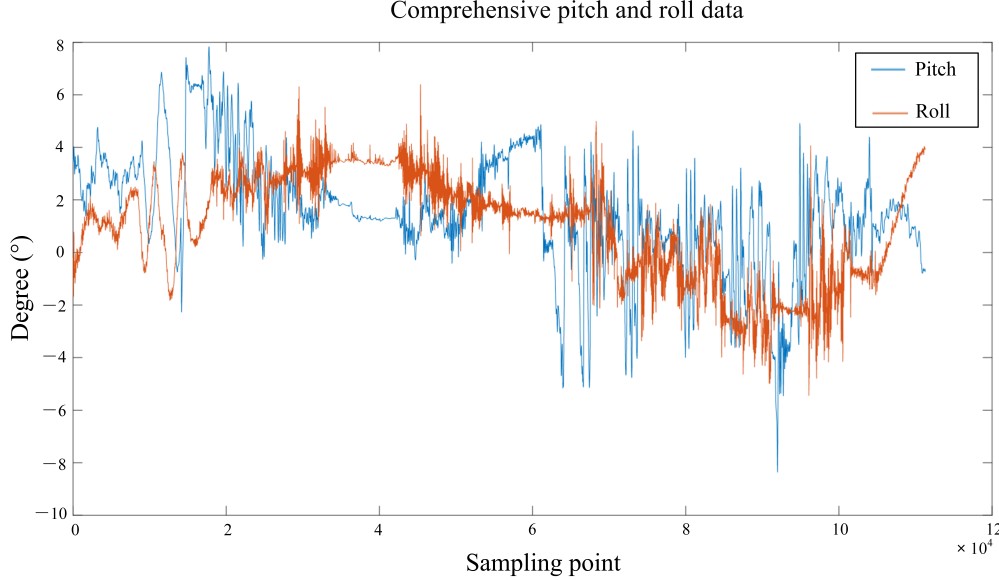

**Figure 14: Comprehensive pitch and roll data recorded by the magnetometer.**



From the data, we can obtain that the sensitivity of the system is less than 0.5 nT as shown in Table 3.

**Table 3.** Comparison between developed system and other main models.

| Indicator | Specific Index | Test Results |
|---|---|---|
| Sensitivity | ≤0.5 nT | After Maneuver Compensation: 0.4157 nT<br>Steady-State Phase: 0.1932 nT |

### 6.3.1. Steady-State Data Processing

Figure 15 presents a comparative analysis between data acquired during the maneuvering phase and the steady-state phase. The figure clearly illustrates that the steady-state phase data exhibits minimal fluctuations and smoother curves, whereas the maneuvering phase data shows significant fluctuations and pronounced undulations. For the steady-state analysis, data from 11:14 AM to 11:17 AM, during which the HOV maintained relatively stable attitudes, was selected. The calculated standard deviation of 0.1932 nT indicates minimal magnetic field fluctuations, thereby confirming the system's stability during this period. In contrast, during the maneuvering phase, substantial variations in the system's attitude resulted in greater fluctuations in the magnetic field data.

Figure 16 provides a detailed comparison between the raw steady-state data and the raw exploratory data. The upper plot in Figure 16 illustrates the raw steady-state data, which exhibits relatively low amplitude with fluctuations within approximately ±0.5 nT. This indicates that the system's attitude was stable during the steady-state phase, resulting in minimal magnetic field data fluctuations. Conversely, the lower plot in Figure 16 depicts the raw exploratory data, which shows significantly higher amplitude values with fluctuations ranging approximately between 46050.2 and 46052.1 nT. This suggests that during the exploratory phase, the system was likely subjected to increased external interferences or significant attitude changes, leading to greater magnetic field data fluctuations.

The pronounced amplitude differences between the raw datasets underscore the substantial impact of different operational phases on magnetic field measurement data. These findings highlight the necessity of accounting for system stability and external interferences when analyzing magnetic field measurements to ensure data accuracy and reliability.





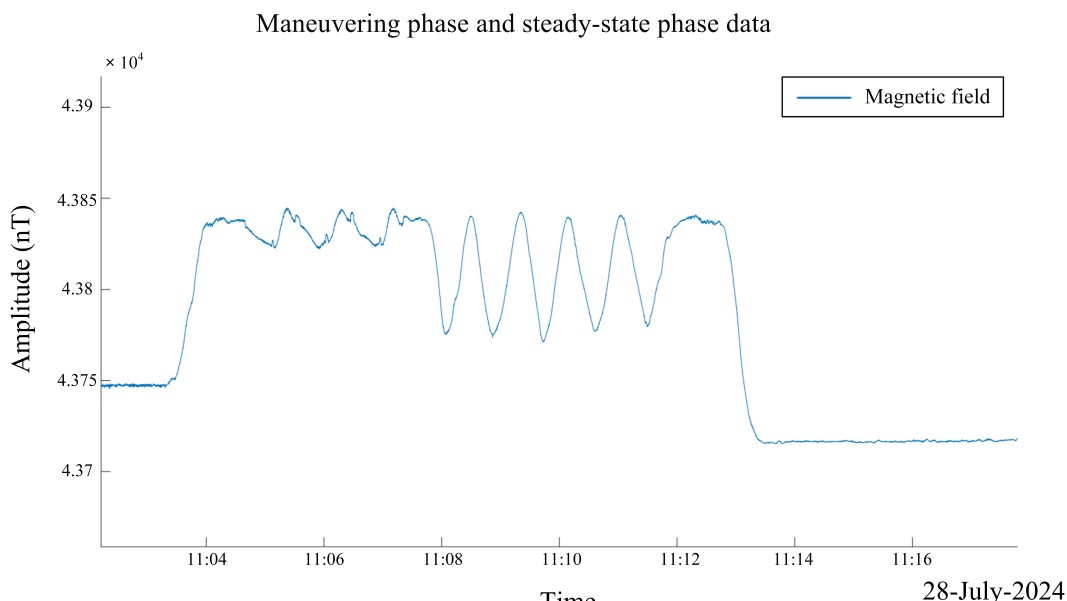

**Figure 15: The maneuvering phase and the steady-state phase data.**

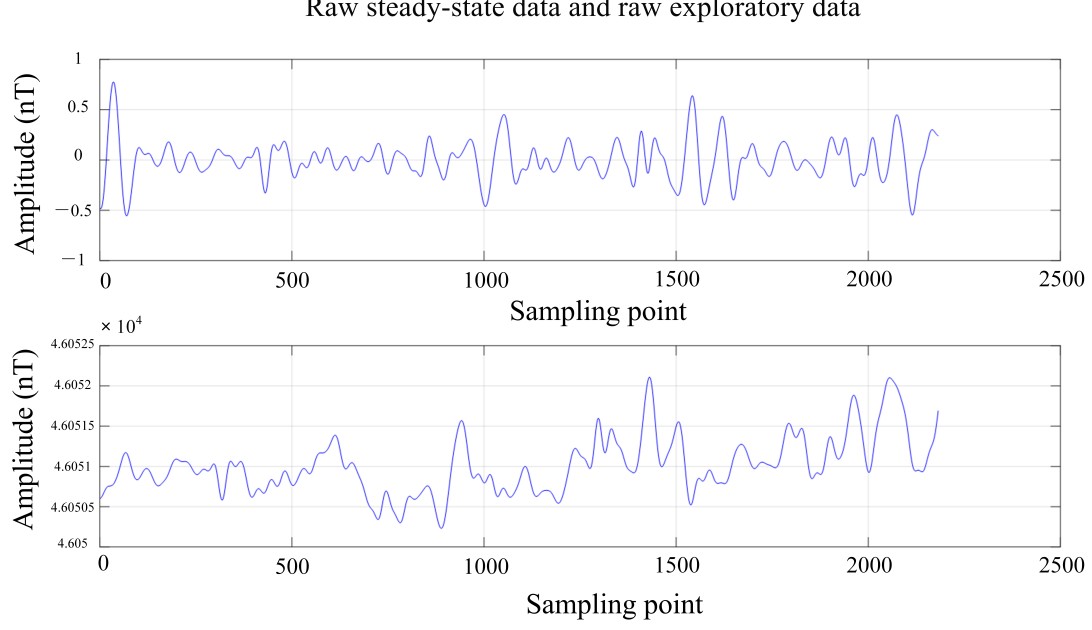


**Figure 16: Comparison of raw steady-state data and raw exploratory data.**





**6.3.2. Magnetic Compensation Data Processing During Maneuvering**

Figure 17 illustrates a comparative analysis of the magnetic measurement data before (blue curve) and after (red curve) the application of the compensation algorithm. The figure clearly demonstrates that the fluctuations in the compensated data are

significantly attenuated, indicating the efficacy of the compensation algorithm in mitigating interference and enhancing data stability. Tables 4 and 5 respectively present the compensation results and the derived compensation coefficients. The improvement ratio for the magnetic compensation is calculated to be 2.13, and the derived compensation coefficients serve as reliable parameters for subsequent data processing. These findings indicate that the compensation algorithm significantly reduces the standard deviation of the measurement data, thereby enhancing the precision and accuracy of the measurements.

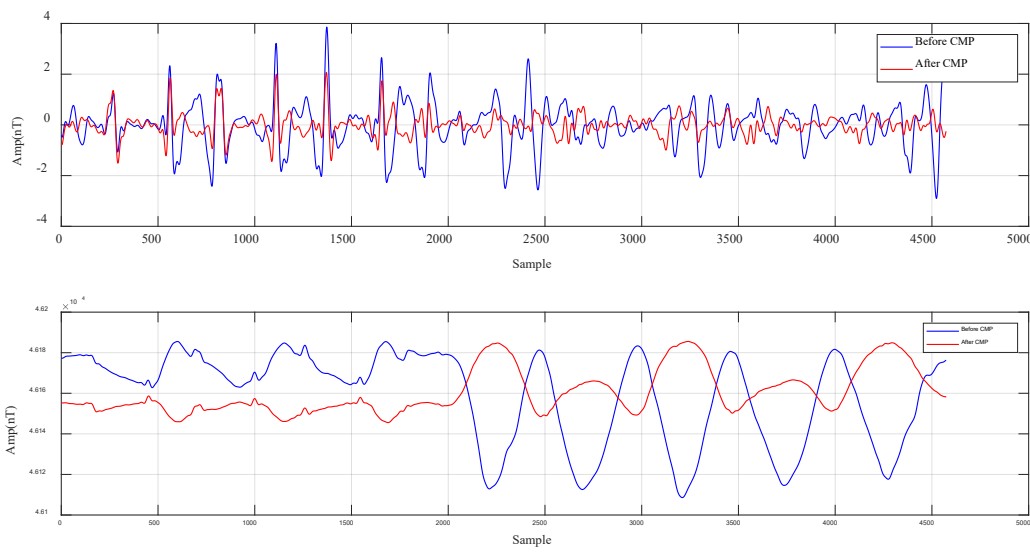


**Figure 17: Comparative analysis of magnetic measurement data before and after compensation.**

**Table 4.** Compensation results.

| Before Compensation (nT) | After Compensation (nT) | Improvement Ratio (IR) |
| --- | --- | --- |
| 0.8870 | 0.4157 | 2.1341 |

**Table 5.** Derived compensation coefficients.



| Index | Compensation Coefficient |
|-------|--------------------------|
| C1 | -3.34400303746229 |
| C2 | -0.0505952353569666 |
| C3 | 5.13813278026741 |
| C4 | 0.0968517738599245 |
| C5 | -0.00459595762233662 |
| C6 | -0.000496776993900312 |
| C7 | 0.0942559884495657 |
| C8 | -0.00157224873261729 |
| C9 | 0.0920085811143404 |
| C10 | -0.00157224873261729 |
| C11 | 0.0920085811143404 |
| C12 | -0.522711782971617 |
| C13 | 0.00362123661798063 |
| C14 | 0.00270405827243547 |
| C15 | -0.00671418980315576 |
| C16 | -0.527221041511662 |
| C17 | -0.00590898328136487 |
| C18 | -0.00116929360761580 |



## 7. Conclusions

This paper introduces a high-precision magnetometer system based on the HOV platform, utilizing cesium optically pumped magnetometers and fluxgate probes for high-accuracy magnetic field detection. The system's software architecture is comprehensively detailed, including key functionalities such as automatic probe switching and magnetic compensation, both of which significantly improve the precision and reliability of the measurements. Furthermore, the upper computer display and control software provides a user-friendly interface for real-time monitoring and system management. Comprehensive laboratory tests and marine experiments conducted on the Shenhai Yongshi platform have demonstrated the system's accuracy and reliability. The probe noise was measured to be as low as 1 pT/√Hz, and the system demonstrated a sensitivity of 0.5 nT. Additionally, the system supports multi-channel data acquisition and incorporates real-time magnetic compensation capabilities. The software has proven to be both user-friendly and stable in performance. Moreover, the integrated software system features real-time monitoring, data recording, data processing, a user interface, and alarm mechanisms, thereby enhancing the system's practicality and reliability. This research provides a robust foundation for the implementation of data acquisition software systems in marine magnetic surveys and exploration.

## Code and data availability

The original contributions presented in the study are included in the article, further inquiries can be directed to the corresponding author.

## Competing interests

The authors declare that they have no conflicts of interest.

## Acknowledgment

This study is supported by the Key Research Program of the Chinese Academy of Sciences (Grant NO.KGFZD-145-22-06-02) and the National Key R&D Program of China (Grant No.2022YFF0706202 and No.2021YFC2801404) and the National Natural Science Foundation of China (Grant NO.42074155).





**Financial support**

This study is supported by the Key Research Program of the Chinese Academy of Sciences (Grant NO.KGFZD-145-22-06-02) and the National Key R&D Program of China (Grant No.2022YFF0706202 and No.2021YFC2801404) and the National Natural Science Foundation of China (Grant NO.42074155).

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
