# Peer review of "Software Program Development of a High-Precision Magnetometer System for Human-occupied Vehicles"

_Geoscientific Instrumentation, Methods and Data Systems, 2024_

## Author Comment (AC1)

We truly appreciate the time and energy you dedicated in carefully reviewing our manuscript.

Your comments were highly helpful. We really appreciate your attention and comments on our manuscript. Our replies are listed as follows:

1. The paper mainly discusses the software, while Section 2.1 introduces the hardware platform. Is the software running on a self-developed hardware platform or a commercially available magnetometer? If it is a self-developed hardware platform, what is its acquisition structure? What are its sensitivity and dynamic range?

    Response 1: The software operates on a self-developed hardware platform. Its acquisition system is designed with an optically pumped probe, a fluxgate probe, a magnetic signal processor, and a high-depth pressure hull. It achieves a system sensitivity of ≤ 0.5 nT, supports a sampling rate of 10 Hz, and offers a dynamic range spanning from 12,165.10 nT to 137,992.85 nT.

2. Please translate the text in the lower right corner of Figure 11 into English.

    Response 2:  We have revised Figure 11.

3. Ensure proper capitalization in Figures 3 and 6.

    Response 3: We have modified Figures 3 and 6 according to your comment.

4. In Table 1, correct the capitalization of "NET communication Interface" and check other tables for similar issues.

    Response 4: We have modified Table 1 and checked other tables according to your comment.

---

## Author Comment (AC2)

We truly appreciate the time and energy you dedicated in carefully reviewing our manuscript. Your comments were highly helpful. We really appreciate your attention and comments on our manuscript. Our replies are listed as follows:

As the title suggests 'Software Program Development,' there is limited insight provided or explanation given about the software and its application. It would be more justified and balanced if these aspects were explained in detail alongside the other content.

Response 1: Thank you for your insightful comment. We appreciate your observation regarding the need for more detailed explanation of the software and its application. In response to your suggestion, we have revised the manuscript accordingly to provide a more comprehensive description of the developed software system. Through these revisions, we believe the manuscript now offers a more balanced and detailed account of the software's development and application, consistently aligning with the title. We sincerely appreciate your valuable input, which has helped us improve the clarity and completeness of the paper.

Line 35-39: How does the system address environmental interference, particularly in cases of natural calamities like thunderstorms? If there is a power surge or electronic failure, how does the system recover, and what role does the software play in ensuring data integrity and functionality?

Response 2: To address environmental disturbances, particularly natural disasters (e.g., thunderstorms), as well as the system's recovery under conditions like power surges or electronic malfunctions, and to ensure the role of software in maintaining data integrity and functionality, the system has implemented the following measures:

Response to Natural Disasters (e.g., Thunderstorms):

1. Electromagnetic Shielding and Filtering: The data acquisition circuit and hardware component housings are designed with robust electromagnetic shielding and filtering capabilities. This ensures that external transient environmental disturbances, such as those caused by thunderstorms, do not significantly impact the system's operational performance or the quality of the data.

2. Abnormal Data Handling: Each frame of raw serial port data undergoes header and tail verification, followed by XOR checksum validation. This prevents corrupted data caused by interference from being transmitted to subsequent stages of data processing. Data that fails the validation is treated as packet loss. For raw serial port data, anomalous data points are filtered out ("spike removal"). Sudden abnormal values caused by events such as engine startup interference or power-on interference from other equipment are constrained within predefined limits. Out-of-range data is replaced with the last valid data point. Serial port data is also monitored for packet loss. If packet loss occurs for less than 5 seconds, the most recent valid data point is used as a substitute to ensure the consistency of sampling rates and prevent time synchronization issues during data processing. If packet loss exceeds 5 seconds, the acquisition system is restarted.

Response to Power Surges and Electronic Failures: The system is equipped with built-in transient voltage suppression (TVS) diodes, which protect electronic components from damage during power surges, such as those triggered by thunderstorms.

We have added that in Line 47-58.

Line 40: What is the latency period for data transfer during real-time magnetic compensation data processing? Could you elaborate on the specific real-time processing functions implemented?

Response 3: Real-time magnetic compensation data processing is executed in the device, with an output frequency of one data packet every 100 milliseconds. Each packet contains 200 bytes of data and can be transmitted simultaneously via the RS422 serial port (baud rate of 115200 bps) and a gigabit network. For data transmission using the RS422 serial port at a baud rate of 115200 bps, the typical delay does not exceed 100 milliseconds. This low latency ensures the system's rapid response to dynamic magnetic field interference, meeting the real-time data processing requirements for applications in fields such as geophysical exploration.

Specific real-time processing functions implemented:

When designing the real-time processing features, we focused on ensuring data integrity, real-time performance, and processing efficiency. The following are the key real-time processing functionalities implemented in the system:

1. Magnetic Interference Compensation Algorithm: The system employs the Tolles-Lawson compensation algorithm along with other optimized filtering methods to correct, in real time, errors caused by interfering magnetic fields in the sensor's operating environment. This ensures accurate measurements of the target magnetic field.

2. Detection and Removal of Abnormal Data Points: By utilizing a custom-designed anomaly detection algorithm, the system marks invalid data points in real time, which may arise from severe external disturbances such as lightning currents or mechanical vibrations, and removes them from the dataset. Simultaneously, it records the timestamps of these anomalies to facilitate subsequent analysis.

3. Real-Time Data Stream Visualization: The software provides real-time graphical visualization of the processed data, allowing users to observe magnetic field variation trends intuitively through a user-friendly graphical interface. The interface supports data zooming and time-specific annotations, enabling quick labeling and preservation of anomalous points identified during the measurement process for further investigation.

4. Real-Time Data Storage and Backup: A high-speed ring buffer storage mechanism has been implemented to ensure that real-time data is not lost due to storage medium delays. Additionally, the system supports periodic backup functionality, which synchronously saves real-time data to an SD card for secure archival.

Figure 1: What is the internet speed required for optimal system performance? Is real-time monitoring conducted over a LAN or WAN? Additionally, is remote access to the system possible?

Response 4: The real-time output data volume for magnetic detection is 16,000 bps, and the system achieves optimal performance when using a gigabit local area network (LAN). Data transmission is limited to the local LAN and does not support remote access.

Section 2.2: Are all the software components mentioned in the paper developed by the authors, or are commercial software applications utilized as well?

Clarification Request: Could you elaborate on the following software components mentioned in the paper?

   Automatic Switching Software

   Data Acquisition Software

   Real-Time Magnetic Compensation Data Processing Software

Data Storage Software

Data Communication Software

If all these software components were developed by the authors, what platforms or programming environments were used? Are there any unique features that distinguish these software systems from existing commercial solutions?

What programming languages or tools were used to develop the automatic switching software?

Can you elaborate on the architecture or specific programming frameworks used for the data acquisition and processing software?

Response 5: Thank you for your comments. The process of the probe automatic switching software: First, the magnetic compensation probe is used to calculate the current magnetic inclination (±90°) and magnetic declination (±180°). Then, combined with the known installation angles of the three optically pumped probes relative to the magnetic compensation probe, the angle between each optically pumped probe and the geomagnetic field is calculated. The specific steps include: obtaining the measurement results from the magnetic compensation probe, determining the installation angles of the optically pumped probes, and calculating the actual angle between each probe and the geomagnetic field through geometric relationships. Ideally, the working angle is optimal when the angle between the optically pumped probe and the geomagnetic field is 45°. Therefore, the next step is to calculate the actual angle between each probe and the geomagnetic field and compare it with 45° to obtain the working deviation value for each probe. By comparing the deviation values of the three probes, the probe with the smallest deviation is selected as the current working probe. In this way, the probe automatic switching software can intelligently select the optimal working probe, avoiding the dead zones of the probes and ensuring continuous and accurate measurements.

Data Acquisition Software and Data Communication Software: The process begins by determining the amount of raw cached data, followed by the retrieval of serial port data. Next, the system performs data verification and judgment to ensure the integrity of the data before proceeding to parse the serial data. It then retrieves the current latitude, longitude, and altitude status from the system's status records. The magnetic field raw data is merged with this location and altitude information. Any packet loss is handled based on packet numbers to ensure data consistency. Finally, the merged magnetic detection raw data is stored into a multi-level cache before the process ends.

Real-Time Magnetic Compensation Data Processing Software: The process starts by attempting to read 18 compensation coefficient files. If the files fail to open, the system defaults to parameter 0 for processing. If successful, the system performs attitude correction for magnetic exploration data. Simultaneously, optically pumped probe data undergoes down-sampling, bandwidth selection, and geomagnetic gradient correction. Magnetic compensation probe data is used to compute an attitude information matrix, which feeds into the attitude correction process. Afterward, the system applies the 18 coefficients for data compensation processing. It outputs the compensated magnetic field data, which is then sent onward, marking the end of the process. Data Storage Software: This process starts with the initialization of the network data input/output module, where the system waits for raw data at 10 Hz. The first-level data buffer receives the data in a loop, distinguishing effective data packets from idle packets. Data parsing and packet loss handling are then performed to analyze XYZ positional data, magnetic field data, and target data, which are passed to the second-level data buffer. The second-level buffer stores processed effective packets in a loop for further extraction. The third-level buffer then receives the cleaned data, extracting and displaying status information

such as device conditions and system functionality. The final stage displays the processed data, including magnetic field visualizations (e.g., time-frequency plots) and target information, ensuring a comprehensive understanding of the system's performance and functionality.

In this work, all core software components mentioned in the paper, including the probe automatic switching software, data acquisition software, and host computer display and control software, as well as the implementation of the compensation algorithm, were fully developed by the authors. These are some of the software modules for the data acquisition and processing software, developed using a hybrid programming approach with C and C++. The software runs on an embedded ARM Linux platform. C language is used to implement functions such as system resource management, thread scheduling, and data communication. Meanwhile, C++ is utilized to handle real-time magnetic compensation data processing and automatic probe switching. The G++ (GNU C++ Compiler) is used as the compiler. One of the key features of this software is its capability to perform magnetic compensation coefficient calculations and real-time magnetic compensation data processing directly on the embedded device.

Table 1 (Page 7): The alignment of "Interface" and "Description" in Table 1 appears inconsistent. Could you clarify or suggest adjustments for better presentation?

Response 6: We appreciate you raising this point. We have modified the presentation.

Dead Zones Mitigation:

How is the optimal working probe determined when multiple probes might provide conflicting data in overlapping regions?

Are there specific algorithms or machine learning models used to predict or mitigate dead zones?

Response 7: Thank you for your comments. When multiple probes operate simultaneously, the system will dynamically select the data from the probe with the optimal working angle in real time as the valid data. The dead-zone issue of optically pumped magnetometer probes is indeed a significant challenge in magnetic field measurements. When the magnetic field direction forms an angle of 90° or 0° with the probe's optical axis, the projection of the magnetic field component on the optical axis becomes too small, significantly reducing the probe's signal sensitivity. To address this issue, we have adopted the following strategies:

1. **Multi-Probe Array Layout**: By deploying a multi-probe array to cover the measurement area, we ensure that even if one probe enters a dead zone, another probe can provide valid measurement data.

2. **Optimal Probe Selection**: For overlapping measurement regions, the system calculates the actual angle between each probe and the geomagnetic field and compares it to 45° (since the ideal working angle of an optically pumped magnetometer probe is 45° relative to the geomagnetic field). The deviation of each probe from this optimal working angle is calculated. By comparing the deviations of all probes, the system selects the probe with the smallest deviation as the active working probe.

Currently, there is no specific algorithm or machine learning model implemented to predict or mitigate the dead-zone issue. However, this is a promising direction for future development.

Hardware Integration:

Are there specific challenges in integrating optically pumped probes at different angles? For instance, does this require precise mechanical alignment or calibration?

Response 8: The installation brackets for multiple probes require high angular precision. To

ensure this, 3D printing technology is utilized to achieve accurate bracket angles. There is a certain degree of overlapping working areas between the probes, ensuring that even with an angular error of less than 1°, no blind spots occur in the operation.

How does the system ensure synchronization between the hardware probes and the software modules?

Response 9: Each probe is equipped with its own signal conditioning circuit, but the data acquisition for multiple hardware probes is handled by a single FPGA. This ensures data synchronization across all probes.

Data Communication and Storage:

What protocols are used for data communication between the data acquisition system and the upper computer main control system?

Can the system handle large-scale data or high-speed data streams without compromising performance?

Response 10: Data communication between the data acquisition system and the host control system utilizes the TCP protocol. The magnetic detection system operates at a data output frequency of 10 Hz, as it does not involve scenarios with large-scale or high-speed data streams.

Communication Protocols:

What measures are in place to ensure reliability and low latency in the RS422-based serial communication and Gigabit Ethernet protocols?

Are there potential limitations in data transmission rates that might impact real-time processing?

Response 11: The magnetic detection system requires a data transmission rate of 2000 bytes per second, which is relatively small. Both RS422 serial communication and Gigabit Ethernet protocols provide very low latency under such conditions, ensuring that real-time processing is not affected. Additionally, CRC verification is employed to ensure data reliability during transmission.

Restart Mechanism:

How does the system ensure that data acquisition resumes smoothly and without duplication after a restart?

Response 12: If data packet loss persists for more than 5 seconds, the data acquisition process will automatically restart. This restart mechanism helps mitigate the impact of transient system errors and ensures continuity of operation. The restart event is also logged in the system's log file for traceability. Frequent restarts of the data acquisition process, however, may indicate underlying issues with the hardware. Troubleshooting hardware problems in such cases is a crucial step to ensuring the stable operation of the system.

Data File Creation:

How does the system ensure that data integrity is maintained when storage space is low?

Are there provisions for automated alerts or logs when data acquisition is halted due to insufficient disk space?

Response 13: The data acquisition and processing software checks the remaining disk space every second. When the available space approaches a critical threshold (default: 200 MB, configurable via a settings file), it automatically deletes the oldest files based on their modification timestamps until the available space exceeds 70%. This automated cleanup process is logged in the system's log file for record-keeping and traceability.

Thank you for your valuable suggestions.

---

## Author Response (AR2)

We have modified introduction and added mentioned references.